# "Komai Nisan Dare, Akwai Wani Online": Social Media and the Emergence of Hausa Neoproverbs

Abdalla Uba Adamu 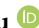

Department of Information and Media Studies, Faculty of Communicaion, Bayero University, Kano 700241, Nigeria; auadamu.ims@buk.edu.ng

**Abstract:** This paper interrogates the changing paradigm in the evolution of traditional African proverbs in the postcolonial setting in which Hausa youth create proverbs centered around the power of both social media and their technologies. In this context, the notion of colonized subjects, cowering under the glare of English linguistic imperialism, is challenged by the Hausa youth through newly fabricated social media proverbs that acknowledge English terms, but use social media platforms to convey what I call 'Hausa technofolk' philosophy. This provides insight into how contemporary African youth force a new narrative in the notion of coloniality.

**Keywords:** Hausa language; northern Nigeria; Facebook; neoproverbs; parallel proverbs; antiproverbs; Gen Z; postproverbials

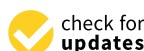



## 1. Introduction

Proverbs are common in all human cultures. Their lack of specific authorship endows them with the significance of folk philosophy, encoding folk wisdom that encapsulates whole ranges of human experiences in science, engineering, technology, leadership, and other aspects of public culture. The proverbs of the Muslim Hausa of northern Nigeria are steeped in mostly Islamic moral philosophy, the Hausa being Muslims since 12th century. Despite the passage of time, Hausa proverbs have remained as constant as the moral situations necessitating their creation.

So far, proverbs have remained straightforward conversational and traditional dialogues, i.e., rooted in folk tradition. However, with increasing urbanization and changing dynamics in social interactivity, proverbs are also changing. These changes, mainly minor alterations to the original proverbs, often add a new meaning to the same proverb. The proverbs that have emerged out of alterations to original proverbs are variously referred to as 'antiproverbs' (Mieder 2007) and 'postproverbials' (Raji-Oyelade 1999).

While these proverb variants are also present in, especially, modern Hausa proverbs, the introduction of modern communication technologies and social media platforms in Hausa societies have contributed to the creation of new forms of proverbs. This is especially true among young users of such platforms that often rework old proverbs or create new proverbs based on the use of such social networks or their underlying communication technologies. I refer to these reworked or completely new proverbs that are based on new media technologies as 'neoproverbs'. I make this distinction right away to link neoproverbs with communication technologies ('karin maganar hanyoyin sadarwa'). New Hausa proverbs not rooted in the technological domain remain that—new proverbs ('sababbin karin magana').

This paper contributes to the debate on the transformation of proverbs in a traditional society by analyzing at least three clusters of modified proverbs in a collection of contemporary Hausa oral culture and paremiology. The first were straightforward 'antiproverbs,' as defined by Mieder and Röhrich (Mieder 2007), or what Raji-Oyelade (1999) labelled 'postproverbials'. The second are what Usman (2014) refer to as 'parallel' proverbs. These

are Hausa proverbs with similar roots and meanings but are applicable to different contexts. While acknowledging these two categories as part of the flow of Hausa paremiology, my main focus is on the emergence of neoproverbs on social media

The paper starts with methodology and then provides the contextual development of proverb alterations in various countries in order to situate the emergence of Hausa neoproverbs within the larger matrix of paremiology. It then presents the emergence of neoproverbs as those located within the social realities of new media technologies.

## 2. Sources and Methodology

This research is qualitative, and does not set out to measure any variables, relying as it does on textual data gathered from online or unidentified individual sources. Although it recognizes critical discourse analysis as the study of written or spoken language in relation to its social context, it does seek out authors or audiences of the texts to critically examine their motives or the context of their usage of the harvested proverbs. This is deliberate, as the approach adopted approximates critical ethnography, which "relies on the qualitative interpretation of data as this examines particular social, cultural, or organizational settings from the perspectives of the participants involved" (Okoko and Prempeh 2023, p. 91).

The data for this paper were harvested from three sources. The first was three authoritative compendiums of traditional Hausa proverbs (Merrick 1905; Kirk-Greene 1966; Yunusa 1978). These provided a corpus of common, traditional proverbs that have retained their classical Hausa folk structure and have remained unmodified. The proverbs documented in these sources serve as a base for comparison with any modifications or the creation of original proverbs.

The second source was community conversational groups in Hausa cities (see, for instance, Youngstedt 2004) These 'hangouts' are often referred to as 'majalisa' (council) in Nigerian Hausa communities, and they serve as a gathering place for conversations and, subsequently, places to hear a variety of proverbs. I have been an active member of two of such groups in the city of Kano, northern Nigeria, since adulthood.

The third source for the data was online forums that I actively participate in, such as Facebook, Instagram, and WhatsApp groups with significant Hausa Gen Z populations, all posting in the Hausa language of northern Nigeria. Indeed, as of November 2022, Facebook alone had dozens of different groups dedicated to Hausa proverbs that provided a rich source of data on neoproverbs and their use among young people in Hausa societies. These online proverbs are categorized into two groups. The first was those that altered existing proverbs using information technology as modifiers. The second are proverbs that do not refer to any existing proverb and are, essentially, new. Their only relationship to existing proverbs is semantic similarity. The ethical issues of this Netnographic method (Kozinets 2002) were addressed by the fact that I was not anonymous in these groups, nor was I actively participating. At the same time, moderators of the groups often requested user-input in discussions of new proverbs. Thus, the sites of my data collection were not bound by privacy or confidentiality concerns.

## 3. Proverbs and Their Transformations

As Mieder (2004, p. 1) pointed out, "of the various verbal folklore genres (i.e., fairy tales, legends, tall tales, jokes, and riddles), proverbs are the most concise but not necessarily the simplest form". Their complexity is reflected in the way they encapsulate social philosophy, particularly for communities that do not have what I may refer to as 'authored philosophies'. Yankah (1989, p. 325) observed that some "scholars like Walter Ong have associated proverb use with nonliterate societies, based on the argument that the mind cannot engage in analytical organization of thought without writing". Yet, as Whiting (1932, p. 278) argued, a proverb is "a short saying of a philosophic nature, of great antiquity, the product of the masses rather than of the classes, constantly applicable, and appealing". Proverbs are contested as reflective of the social philosophies of their communities, not just quaint 'non-literate' societies.

Further, Hatipoğlu and Daşkın (2020) suggested that proverbs have been defined in various ways. There is the structural approach, which describes proverbs as propositional statements including at least a topic and a comment (Dundes 1975; Milner 1971). Others approach the study of proverbs from ethnographic and cultural perspectives, which see proverbs as typically spoken, conversational forms whose sources are not known and which usually have a didactic function (Giddy 2012; Norrick 1986). Still, others prefer to follow an empirical approach, which helps them study the modifications to proverbs (Mieder 2004). However, I adopt Wolfgang Mieder's definition that "a proverb is a concise statement of an apparent truth which has, had, or will have currency among the folk" (Mieder 2004, p. 4).

The cultural significance of proverbs is often reflected in its community value as a community memory. African societies conduct the vast majority of their social discourse through oral literature, and proverbs provide a very easy outlet for this. Proverbs express thoughts effortlessly and encapsulate social and communal philosophies. They also reflect life's truisms and serve as a moral compass. As Kirk-Greene pointed out,

> [p]roverbs enshrine much of the cultural heritage of a people—their traditions, their history, their wisdom and their ethics. More than this, in the absence of a rigorous written literature, they may serve as the guardian and carrier of a nation's philosophy and genesis. (Kirk-Greene 1966, pp. ix–x)

Within this context, proverbs in Africa acquire a high social value that elevates them to an art form. Thus, according to Raji-Oyelade (2012, p. 27),

> Major scholarships on African proverbs have sustained the idea of the sacrosanct structure of the proverb text. I want to argue that the notion of the fixity of form is almost contradictory to the original idea of the dynamisms of societies and cultures, that it is impossible not to recognize a certain radical shift or the transgressive force in the making and use of proverbs in recent times.

Perhaps because of their rootedness in folklore, the historiography of communities, and social dynamism, Litovkina et al. (2021, p. v) cautioned that

> [p]roverbs are by no means fossilized texts but adapt to different times and changed values. While antiproverbs can be considered as variants of older proverbs, they can also become new proverbs reflecting a more modern worldview. In Europe and North America, the genre of transformed proverbs is becoming ever more popular, especially due to the mass media and the Internet.

The changes to proverbs provide a fascinating example of transformation in oral literature and also reflect the dynamism of language and discourse. In Africa, these transformations reflect a new shift towards postcolonial oral discourse. As Akinsete (2019, pp. 240–41) argued,

> with the fast-changing modern African society in view, the interrogation of the postcolonial has continued to generate intriguing perspectives. The effervescent discourse of decolonization, as expressed in arrays of literatures, divulges among other issues the rhetoric of change/adaptation in postcolonial African space.

Structurally, proverbs as complex sentences are made up of a topic and a comment. These translate as

> one clause and one or more subclauses; the subclauses may be adjectival, nominal, or adverbial. The structural balance in these proverbs is asymmetrical, with the subclause being dependant on the main clause (Coinnigh 2015, p. 114)

In the Hausa bipartite structure of proverbs (Jang 2002), the subclause would not stand on its own and has no discernible meaning outside of its relationship to the main clause. Two different terms have been given in paremiology for the alteration of the subclause in a proverb.

The first was by Wolfgang Mieder and his colleague Lutz Röhrich (Mieder 2007), who coined the German expression 'Antisprichwort' (antiproverb) for this alteration, and

popularized it as a general label for such innovative alteration of and reactions to traditional proverbs (Litovkina 2015). Mieder explained antiproverbs succinctly as "parodied, twisted, or fractured proverbs that reveal humorous or satirical speech play with traditional proverbial wisdom" (Mieder 2004, p. 28). He further pointed out that:

> the conscious manipulation of so-called fixed proverbs is absolutely nothing new. After all, proverbs are anything but sacrosanct pieces of universal wisdom. Instead, they express generalized observations and experiences that are as varied as life itself. (Mieder 2007, p. 17)

The second term for altered proverbs was by Raji-Oyelade (1999, p. 75), who argued that the "supplementary proverb, the product of which may be inadvert or unintentional, is what I call the phenomenon of the *postproverbial*" (emphasis added). He further held that the postproverbial is "situated in the subfield of transgressive paremiology . . . of alternate proverbs which are radical and parallel compositions instead of conventionally accepted and given proverbs in traditional societies" (Raji-Oyelade 2012, p. 27). It would appear from the various studies available that postproverbial, as a term, was adopted not only by Raji-Oyelade but also by other African paremiologists (see, for instance, *Matatu*, 52).

However, neither Mieder nor Raji-Oyelade considered the status of newly formulated proverbs, rather their interest was on the alteration of existing ones, especially those infused with social media referents. Thus, I coin the term 'neoproverb' for the category of proverbs that have their origins wholly in internet technology.

Further, besides the use of different terms, it is not clear what differentiates an *antiproverb* from a *postproverb*, except that the latter is more applicable to African proverbs and tied up with the notion of postcoloniality. As Raji-Oyelade and Oyeleye (2019, p. 232) noted,

> postproverbiality is purposively named and situated within a string of 'posts'. As such, postproverbials are postcolonial, poststructural, postmodern, linguistic reinventions. They are postcolonial in the African/Third world context since their use and popularity became noticeable post-colonial. They are poststructural because they stand at variance with the norm and are responses to a pre-existing structure—proverbs, whose secure sense of meaning they disrupt. The postproverbial is postmodern because it seeks to create 'a new from the old' through the use of playful parody and pastiche, which, as an imitation of another's style, prompts us that the text is not "original" but constructed.

Thus, Raji-Oyelade and Oyeleye's interpretation of postproverbials as "playful parodies of the once sacrosanct African proverbs" (Raji-Oyelade and Oyeleye 2019, p. 232) integrates the very concept of postproverbials exclusively with African paremiology. This is confirmed with the development of the Postproverbials in African Cultural Expressions (PACE) internet site, which describes itself as a "trans-national initiative which brings African scholars in the humanities together with the ultimate aim of contributing to the body of modernist and radical proverbs which are created mainly in urban communities almost all over the African continent". The initiative was supported by the Institute of Asian and African Studies, Humboldt University of Berlin, Germany. The PACE site contains Postproverbials in Efik, Fulɓe, Hausa, Mwaghavul, Yoruba, Igbo, Ibibio (Nigeria), Gikuyu (Kenya), Akan, Kasem (Ghana), Kiswahili (East Africa), Luo (Kenya), and Shona (Zimbabwe). Perhaps not surprisingly, the site was curated by Raji-Oyelade (PACE 2022).

Indeed, this term is rarely used by non-African paremiologists. Yet, Iyabode Daniel disagreed with the notion of 'blasphemy' that Raji-Oyelade's 'postproverbial' theory suggests is implied in a proverb. She points out that

> in the age of smartphones and tablets that are handheld and not to be swallowed, the wisdom of the present age has moved from the elders to the young. One can argue that this is a kind of wisdom and not necessarily cultural wisdom. That may appear true. However, we should not forget that cultural wisdom also has to do with cultural dynamism. (Daniel 2016, p. 70)

Similarly, Okhuosi (2020, p. 380) argued that, although postproverbiality is a development from proverbiality, this transition

> is not limited to the semantic sense of time only; it is also the radical revisions of conventional proverbs among the youths. It is characterized by wit, humour, modernisation, ideological radicalism. It draws heavily from the syntax and semantics of the wealth of traditional proverbs but reproduces them through the deconstruction and infusion of modern ideologies.

However, according to Raji-Oyelade (2013, p. 15), "postproverbials are translatable in Yoruba as *áṣákaṣá*, that is, the dynamic act of the cultural deviant, the prodigal text which always attempts to overwrite its own source". Similarly, Yakasai (2021, p. 16), in his analysis of social media proverbs, noted that

> [a]lthough the morphology, structure and semantics of the vast online corpus of Hausa idioms and proverbs have been severely damaged (*language pollution*), it can be addressed by making and implementing language policy. In short, while the top-down language policies can help protect the normative aspects of language, the bottom-up language use shows the language characteristics of young netizens, which could be studied as a language phenomenon (emphasis added)

Both Iyabode Daniel and Ronke Okhuosi have introduced a dimension that deconstructs Raji-Oyelade's postproverbials as 'blasphemies' or as Yakasai's 'language pollution' (p. 17). While allowing the term the dignity of its existence and applicability, as, for instance, demonstrated in a whole issue of *Matatu*, which was devoted to various articles on postproverbiality in Igbo (Abana and Eke 2019), Fulɓe (Raji-Oyelade and Ango 2019), and Arabic (Akewula 2019), it is clear that its source as a label for modified African proverbs is in contention.

Thus, Wolfgang Mieder and Adeyemi Raji-Oyelade force us to take a second look at the evolving nature of proverbs and the way the current social culture deconstructs them. It might be interesting to excavate further back to find that what we now see as deconstructions of what might be considered 'classic' folk philosophy indeed also evolved from a similar deconstruction of an earlier folkloric thought. Thus, in looking at the various iterations of proverbs, Mieder (2007, p. 18) cautions that

> [i]t is not enough to identify hundreds of antiproverbs and place them into collections organized according to the original proverbs followed by the altered texts or thematically by the subjects and meanings of the antiproverbs. Scholars must also interpret the use and function of anti-proverbs in oral and written contexts and reflect upon the significance of this preoccupation with antiproverbs by the folk themselves.

From these explanations about what constitute an antiproverb, what emerges is that an antiproverb must be based on a known proverb to have full effect. To illustrate this, Litovkina et al. (2007) and Mandziuk (2016) collected a series of antiproverbs from various sources, including literature, newspapers, films, and the internet. A few of these included the following:

*Classic*: Man proposes, God disposes.

*Antiproverb*: Man proposes, his mother-in-law opposes

*Classic*: A friend in need is a friend indeed

*Antiproverb*: A friend that isn't in need is a friend indeed

*Classic*: What can't be cured must be endured

*Antiproverb*: What can't be cured must be insured

Each of these altered proverbs indicates a contemporary narrative that reimagines the original proverb to suit current perceptions of social culture. Similar perceptions are found

in the Igbo language proverbs of south-eastern Nigeria. Egbara (2020) illustrated a few examples, with his own translations, although he preferred to use the term postproverbials instead of antiproverbs:

> *Proverb*: He who fetches ant-infested firewood has only extended an invitation to the lizards
>
> *Postproverb*: He who fetches ant-infested firewood has fetched a double problem.
>
> *Proverb*: When a child washes his hands clean, he eats with kings
>
> *Postproverb*: If a child makes money, he eats with kings
>
> *Proverb*: Whatever a man sows, that he will reap
>
> *Postproverb*: Whatever a man sows, he enjoys

The social culture of the Igbo is reflected in these examples. For instance, the proverb about cleanliness and eating with kings suddenly became a contextual proverb that reflects wealth, and thus glorifies material acquisition higher than the dignity of sharing a meal with a revered elder. Enjoying what one sows is a syntactical variation of reaping what one sows.

## 4. Modification of Hausa Proverbs

So far there has not been a detailed study of Hausa antiproverbs, either in literary texts or on social media. The nearest was by Usman (2014), who studied what she called 'parallel' proverbs. However, these were Hausa proverbs with similar roots and meanings but were applicable to different contexts. Thus, she quoted parallels as "elements of equal importance which are expressed in similar grammatical form" (Usman 2014, p. 893). For instance, she gave an example of a parallel proverb as the following (my translations):

> *Proverb*: In ka ga raƙumi, ka ga Buzu/wherever there is a camel, there is a Tuareg
>
> *Parallel*: In ka ga wata, ka ga Zara/wherever you see the moon, you will see Sirius

In the first proverb, 'Buzu' was an onomatopoeic appropriation of the word 'buzuzu,' which refers to the common variety of dung-beetle (*Scarabaeus satyrus*, which makes buzzy angry noises when disturbed), a term thus indicating a short-tempered person. Since the Tuareg (Berber-speaking pastoralists in the West African Sahara) were perceived by the Hausa as being easily provoked and short-tempered, the shortened term, *Buzu*, came to refer to them. Camels are used as a form of transport across sand dunes by the Tuaregs throughout the Sahara, and thus, they came to symbolize each other.

In the second example, a similar juxtaposition is involved where pairings are noted. The coupling here refers to the moon appearing almost at the same time as the dog star, or Sirius. The Muslim Hausa refer to Sirius as Zahra, which means 'bright'. A variation of the term is also used a female name.

As interesting as these parallel proverbs are, they nevertheless represent contextual, different circumstances that have led to similar observations. They are not subversive, or as Raji-Oyelade would describe, blasphemous, since they are only intertextual to the originals. These differ from true Hausa antiproverbs.

### 4.1. Hausa Antiproverbs

A collection of Hausa antiproverbs collected from interaction with people at community hangouts (referred to as 'majalisa' in most Hausa cities) in the city of Kano, northern Nigeria, reveals that they use substitution devices, just like other antiproverbs. In so doing, they modify, extend, and often amplify the original proverb by providing a new, often contemporary narrative meaning to the base proverb. For this device to work, the antiproverb subclause must be connected to the main clause of the old one. Below are five examples that show the universal character of antiproverbs:

> *#1: Proverb*: In kaga gemun ɗan uwanka ya kama wuta, shafawa naka ruwa/if your neighbor's beard catches fire, wet yours

*Antiproverb*: In kaga gemun ɗan uwanka ya kama wuta, aske naka/if your neighbor's beard catches fire, shave off yours

This 'classic' proverb about beards is both antiproverb and also seen as antisocial, because the proverb seems to encourage selfishness—instead of rushing to help a neighbor in distress, the protagonist is busy protecting himself, thus categorizing the proverb as what Doyle (1972) called a *counter-proverb*, because of its negation. The antiproverb, then, is a deeper immersion of the same cautionary behavior reflected in the original.

*#2: Proverb*: Da tsohuwar zuma, ake magani/aged honey has best medical properties

*Antiproverb*: Da tsohuwar zuma gara sabuwa/rather a new honey than an aged one

This proverb is derived from an Islamic description of the properties of honey in the Qur'an (Al-Ghazal 2013). Honey is generally used by Hausa Muslims in most variations of alternative medicine. The antiproverb reflects the same philosophy but in a new social terrain—accepting the curative powers of honey, but believing a newly produced honey is better than an aged one that might have been contaminated or rendered impure. The proverb and the antiproverb both challenge the widely held belief that honey does not 'spoil' no matter how aged, especially if not mixed with some preservatives (that necessitate expiration dates in modern manufacturing). Unlike wine that is said to get better with age, they consider the healing potency of honey as being weakened by age.

*#3: Proverb*: Komai nisan jifa, ƙasa zai dawo/whatever goes up in the sky, will come down

*Antiproverb*: Komai nisan jifa ba zai taɓa sama ba/whatever goes up, will never touch the sky

Many proverbs in all societies often have scientific connotations. The third example of Hausa antiproverbs demonstrates an understanding of the laws of gravity—what goes up, comes down. Yet the antiproverb does not blaspheme the original in suggesting that no matter how high a stone is thrown up, it will never touch the sky; indeed, it reinforces the pull of gravity (objects will never touch the sky, but will fall back to the ground), confirming the validity of the original proverb. While the original proverb (also found in other societies) is also a reminder that good things will not last forever, in the Hausa ethnographic context, it keeps individuals rooted to their social realities no matter how successful they become.

*#4: Proverb*: Kowa ya ci zomo, ya ci gudu/whoever eats a rabbit, will surely be a sprinter

*Antiproverb*: Kowa ya ci zomo, ya ci daɗi/whoever eats a rabbit, will have a delicious meal

Rabbits are excellent sprinters, swiftly disappearing into their warrens, giving the impression in the Hausa proverb that whoever eats rabbit will acquire the same property (#4) of speed. The antiproverb is not convinced of the connection between eating a rabbit and becoming a sprinter. Indeed, no one is known to have acquired extra sprinting abilities by simply eating a rabbit. Thus, the antiproverb focuses more on the culinary property of the rabbit as a rare meat dish, by ignoring how it was caught.

*#5: Proverb*: Bayan wuya sai daɗi/relief follows difficulties

*Antiproverb*: Bayan wuya sai haƙuri/patience follows difficulties

Relief is expected as a reward that follows suffering (#5). This proverb is based on two Qur'anic verses: "Fa Inna ma al usri Yusra. Inna ma al usri Yusra"/"So surely ease (comes) with every hardship. Verily, with (this) hardship (too) there is ease" (Qur'an 94: 5, 6). The overall effect of the proverb is to teach patience—a fact captured in the substitution with the insertion of 'patience' in the antiproverb. This particular substituted antiproverb has a recent and political origin, as it was coined in northern Nigeria to cope with perceived

hardship in the regime of a once popular President whose policies were believed to have made life difficult for the masses throughout his presidency, which lasted from 2015 to 2023.

In each of these examples, the subclause of the classic proverb encapsulating traditional Hausa oral literature is substituted to reflect a wholly different philosophy, often serving as antiproverb.

### 4.2. Hausa Neoproverbs

Hausa neoproverbs reflect the thinking of the Hausa Gen Z social cluster of the online community of Hausa speakers. This group became the source of neoproverbs as a result of their rapid embrace of urban life, modernity, and, indeed, postmodernity in all aspects of their lives. They are postcolonial without having experienced the colonial. Mostly born between 1997 and 2012, they occupy the same demographic as the global Gen Z youth cluster. According to Katz et al. (2021, p. 7):

> Gen Zers, also called postmillennials, Zoomers, or iGen-ers, are the first generation never to know the world without the internet. The oldest Gen Zers, now in their mid-twenties, were born around the time the World Wide Web made its public debut in 1995. They are therefore the first generation to have grown up only knowing the world with the possibility of endless information and infinite connectivity of the digital age.

As 'digital natives,' they are cloistered in online communities with little human conversational interactions. This closed interpersonal space has at least two effects: first it creates a new linguistic ecology for them, and second, it shields them from what is 'normal,' enabling them to create their own new, normative behaviors. For this group, the availability of technology is integral to their lives and shapes their practices, languages, thoughts as well as their perceptions of public culture. They are, therefore, rich sources of proverbs that often have no antecedents, and they have created—without knowing it—what can be traditionally considered 'blasphemous' neoproverbs.

All the Hausa neoproverbs appraised in this study gravitate toward the users' understanding and appreciation of and expertise in online communication. For these users, the online forums and the communities formed there have become new social media village squares, where people hang out and share perspectives. Online words and expressions have become part of their everyday lexicon. Perceptively, they pass judgements on the various devices they use to communicate, particularly phones. Such devices have also evolved into status symbols.

Interestingly, this cohort of online users intensively employ code switching in their construction of neoproverbs, incorporating English as medium of expressing their newly created proverbs. This, indeed, adds a new dimension to the emergent neoproverbs, which have become necessary for them as a result of a lack of local translated equivalents for social media and communication technology terms.

The neoproverbs in this study, which were collected from Facebook, conversational, and WhatsApp groups, as indicated earlier, are divided into three categories. The first are those that approximate antiproverbs (or postproverbials) in that, while their main clause is often rooted in a traditional proverb, their subclauses are firmly located within social media usage. Despite their traditional clauses, I prefer to name them neoproverbs because their subclauses are not modifications of the existing subclause of the proverb since there is no negation, but rather a completely new thought. The second are those wholly rooted in social media usage. The third are those in which both the clause and subclause are dependent on a communication device, particularly smartphones. The second and third groups have no antecedent origins in any prior proverb structure, as both the clause and subclause are wholly new creations.

*4.3. Neoproverbs with Social Media Clauses*

These are neoproverbs with a traditionally structured proverb clause, followed by a subclause based on social media, thus earning the 'neo' affixation. Examples include:

*#1: Proverb*: Wanda ya riga barci, shi zai riga tashi/Early to sleep, early to rise

*Neoproverb*: Wanda ya riga log in, shi zai riga logout/Early to log in, early to log out.

The concept of 'log' in and 'log' out in the subclause is derived from the internet time purchased by young people at internet cafes in northern Nigerian cities. The time purchased was usually hourly, with a timer counting down to the end of the funds allocated. In a competitive spirit among those wishing to log in earlier, the proverb reminds them that they will also log out earlier. While the folk proverb clause is referring to the healthy benefits of sound sleep, the neoproverb harks at the curtailment of enjoying internet services.

*#2: Proverb*: Duk abin da ruwan zafi ya dafa, in aka yi haƙuri, ruwan sanyi ma zai dafa/Whatever hot water boils, will also be boiled by cold water, if you are patient enough

*Neoproverb*: Duk wayar da za ta yi Instagram, za ta yi WhatsApp/Any mobile phone that can run Instagram, can also run WhatsApp

The folk proverb is about the virtues of patience, and the same philosophy is conveyed by users of mobile phones concerning the trending applications used. When Instagram became available in 2010, users of WhatsApp, which became available in 2009, felt that the latter application would not run on their mobile phones. This was clearly ill-informed by the perception that, as a result of the focus of Instagram on photographs, mobile phones running Instagram would not run the predominantly text-based WhatsApp. This reveals a lack of understanding regarding the dividing line between social media and social networks. While Instagram, with its emphasis on images and videos, is categorized as social media, WhatsApp, with its focus on conversations, is essentially a social network.

*#3: Proverb*: Gaba da gabanta, aljani ya taka wuta/There is always a higher power, a spirit being has stepped into fire

*Neoproverb*: Gaba da gabanta, an yi hacking account ɗin ɗan Yahoo/There is always a higher power, the account of a scammer has been hacked

Spirit beings (Islamically, jinns, intelligent spirits of lower rank than angels and created out of the essence of fire) should not feel the heat of the fire; but when they do, things are decidedly serious. The folk proverb is drawing attention to the fact that no matter how large or intimidating a person or thing is, there is likely to be an even larger or more intimidating person or thing somewhere. The neoproverb captures this fairly with its reference to what in Nigeria are referred to as 'Yahoo boys' ('ɗan Yahoo,' in Hausa). These are usually young, male scammers and fraudsters adept at using the internet to extort millions from innocent citizens, especially in European countries, with false tales of either romance or economic investment (see Mufutau 2020). Hacking the account of a hacker captures the essence of 'there is always a bigger fish in the sea'.

*#4: Proverb*: Komai nisan dare, gari zai waye/No matter how deep the night, there will be dawn

*Neoproverb*: Komai nisan dare, sai ka samu wani online/No matter how late the night, there is always someone online

While the folk version is about raising hopes—there is always light at the end of the tunnel—the neoproverb reflects an ethnographic reality of online addiction: there are always online users, a fact not lost on parents of Gen Z children.

*#5: Proverb*: Tsintacciyar mage, bata mage/A stray cat is never a good cat

*Neoproverb*: Tsintacciyar budurwa a Facebook, ba ta ƙarko/A Facebook girlfriend does not last

While cats are as adored in Hausa communities as in any other community across the world, there is a preference for known 'purebred' and fully house-trained cats. A stray cat could come with all sorts of unknowns, thus the folk proverb indicating a lack of trust for such stray cats. The neoproverb extends this notion to online Facebook friendships because Hausa communities prefer 'girl next door' romantic relationships with someone whose 'pedigree' is known. The neoproverb, therefore, warns against a 'stray' girl picked up on Facebooks as an unknown entity with unpredictable behaviors. However, in a reply, girls also came up with their own proverb that counters this, as I will show later.

*4.4. Neoproverbs with Clauses and Subclauses Based on Social Media*

The second dataset contains neoproverbs in which both the clauses and subclauses are based on social media and have no antecedents, as was the case in the first group. Examples include:

*#1:* "Salam, good night", in ji sabon ɗan Facebook/Peace on you, says new entrant to Facebook.

Typical of Muslim communities, every entrant into a new social culture begins with a 'peace upon you' ('As-salāmu ʿalaykum,' shorted to 'Salam'). For many young Hausa having just acquired a social media account, particularly Facebook or WhatsApp, the usual greeting is 'Salam' before engagement with any thread of discussion. The greeting shows decorum, respect, and moral upbringing—even if the thread being discussed is somewhat less than desirable to the Hausa cultural universe. Jumping directly into a thread without the greeting is considered bad manners. A newly arrived member of any social media platform is, therefore, clearly visible by the way they enthusiastically announce their presence with the 'salam' greeting, and they are often the object of derision of being a simpleton who has just arrived to the social media world. In this, the Hausa online culture simply extends to in-person public culture.

*#2:* An yi ba a yi ba, an daina Facebook, an koma WhatsApp/the same old thing, quitting Facebook for WhatsApp.

When Facebook made inroads among Hausa communities, its potential for enabling free expression was frowned upon by the Muslim conservative establishment. Facebook was seen as being the home of all that was undesirable in Islam—despite many pages and groups promoting Islam. As a result, parents and community leaders started discouraging young people from using Facebook. A few years later, WhatsApp became available. Based on the personal space of an individual's phone, it offers more privacy and opportunities to engage in whatever behavior one wanted, including those frowned upon on the more public Facebook. Thus, the neoproverb is pointing out that 'the song remains the same'.

*#3:* A yi dai mu gani, soyayyar soshiyal midiya/As if it'd last, social media romance.

This references the emerging romance on social media platforms as a counterculture. The traditional pattern of Hausa courtship is based on in-person interaction between a boy and a girl at the girl's family house and always chaperoned by a younger female sibling of the girl, or at least in a semi-public space. This is to discourage impropriety, especially with overcharged hormones kicking in.

Before assumptions of sexual awareness made possible by entertainment media engagement, boys and girls courted each other at night, again at the girls' house. Many Muslim states in northern Nigeria issued fatwas (Islamic rulings) prohibiting such night courtship when a series of unwanted, out-of-wedlock babies started to appear in many households. An Islamic moral police force (hisbah), where existing, was empowered to stop night courtship and shift it, if necessary, to late afternoon after the last daytime prayer.

When social media became available, courtship shifted online, but participants felt it lacked the soul and authenticity of in-person courtship, as speech nuances and body language, essential in Hausa non-verbal communication and indicators of true affection, cannot be judged. The general feeling among a few Gen Z social media citizens engaged in

this topic believe that such social media romances rarely last, as the pictures used as avatars are carefully composed to attract romantic partners and, therefore, do not reflect the true nature of the owner. Thus, this proverb casts doubt on the authenticity of online courtship.

*#4:* Tsit ka keji, uwar gulma tayi cikin shege a social media/You could hear a pin drop on social media, a gossipy woman has an unwanted pregnancy.

"Uwar gulma" is a gossipy woman who cross-pollinates gossip from one person to another, and is always broadcasting bad news about other people, especially on the social media. When she becomes pregnant out of wedlock, her social media handles become completely silent. The metaphor also refers to the extremely textually vocal nature of social media messaging, where there is always someone saying something, no matter how late the hour.

*#5:* Aikin kenan, a chatting, budurwa ta gama Sakandare/Constantly at it, nothing but chatting, a young girl has finished Secondary School.

This reflects a sense of freedom for mainly young Muslim girls who are usually thought of as best 'seen, not heard'. Hausa girls usually attend a highly regulated high school, with little conversation and certainly no phones or other forms of technical communication. The proverb, therefore, captures the celebration of vocal freedom when a young lady finishes high school (usually at 18 years) and, subsequently, has all the freedom she craves through chat rooms. 4.6.

### 4.5. Neoproverbs Based on Smartphones

These are neoproverbs with a general focus on telephony, but with specific reference to and preference for smartphones. In these neoproverbs, neither the clause nor subclause are dependent on any prior folk proverb, though they borrow the general syntactic structure of a typical Hausa proverb. They are rooted in the technologies of mass communication. Both Mieder (2007) and Konstantinova (2014) covered proverbs on mass communication, but in different ways, and certainly different from my conception of neoproverbs as rooted in communication technologies.

For instance, while Mieder (2007) has discussed proverbs based on mass communication, his samples were what could be referred to as substituted proverbs, since they only shared onomatopoeic structure with an 'original' proverb—nothing actually linked the original with the substituted, especially for those unaware of the originals. Mieder was dismissive of these 'anti-proverbs,' suggesting that "there is no doubt that most anti-proverbs are one-day-wonders in that they will never enter general folk speech by gaining a certain currency and traditionality" (p. 17). Examples include Mieder (2007, p. 21):

The medium is the message/The modem is the message

Pennywise, pound foolish/Pentium wise; pen and paper foolish

Beware of Greeks bearing gifts/Beware of geeks bearing GIF's

The meek shall inherit the earth/The geek shall inherit the earth

Don't bite more than you can chew/Don't byte off more than you can view

It could be argued that, strictly speaking, these are idioms—often cryptic phrases with their own meaning—rather than pure proverbs, which have advice to another person ingrained in them. Even at that, the anti-proverbial nature of these examples is located in a totally different social culture from that of the 'pure' idiom. Thus, despite their being rooted in technology, I do not accept these examples (and similar ones along these lines) as neoproverbs—perhaps they are neo-idioms. I make this distinction to further clarify the nature of what I regard as neoproverbs. On the other hand, Konstantinova (2014) merely quotes various characters in news reels, TV shows, and journalists' reports using proverbs, thus focusing on the frequency of the utilization of proverbs in mass media discourse.

The first mobile phones available to Hausa youth in urban clusters were cheap ones from lower end Chinese companies. An early affordable model was from Vivo Communication Technology Co. Ltd., with their phones being marketed and stylized as vivo™. The

low-end phone market was soon joined by Tecno™. Although both companies eventually marketed smartphones with full android operating systems, their earlier models used Symbian operating systems.

The iPhone then, and still now, being the highest end smartphone was affordable only to the extremely wealthy. Children of such a class usually end up with lower versions of the iPhone when their parents or relatives move to a newer model. The iPhone has thus become a status symbol and is enviously looked upon by owners of cheaper phones. When Google introduced the Android operating system for other phones, the youth who did not have the financial means to upgrade to an Android phone were stuck with Symbian OS phones. They became the butt of new social media technology proverbs. Such proverbs serve as both neoproverbs as well as a flouting of class differences. Examples include:

*#1:* Zagi ai ba ya ƙari, wai an ce da mai vivo™ buɗe iCloud/Insults don't result in a tumor, a vivo™ owner has been challenged to open an iCloud account.

Since the early Symbian vivo™ phones had no iCloud facilities, users of iPhones are flaunting their iPhone's capabilities for such services. iCloud, launched in 2011, is a service from Apple (the manufacturer of iPhones) that securely stores users' photos, files, notes, passwords, and other data in electronic 'cloud' storage. The iCloud has become critical to phone users who do not have any means of storing the increasingly large numbers of pictures they take with their phones. A variation of elitism in phone ownership is a neoproverb, which states the following:

*#2:* Abin na manya ne, mai vivo™ ya ga mai iPhone ya na ɗaga selfie/These are the true elites, a vivo™ owner on seeing an iPhone owner taking a selfie.

The self-portrait taken with a smartphone camera also has also come to represent ultimate urban cool and modernity among Hausa youth, especially those with iPhones and before the advent of lower-end smartphones. Their ability to take personal photographs in various locations performing various acts has become another status symbol, something owners of lower-end phones have no access to. The proverb, therefore, reflects such class division.

*#3:* Komai daɗin 2go, bai kai ya WhatsApp ba/No matter how good 2go is, it is not a touch on WhatsApp.

The app 2go is one of the few African mobile messenger social network services that caught the attention of many young people in Africa when it first appeared in 2008. When WhatsApp appeared a few years later, it replaced 2go as the most popular social networking application for mobile phones. Despite its popularity, though, there are those who remained loyal to 2go and refused to switch to WhatsApp. This proverb is a celebration of freedom from 2go, for despite its familiarity, the protagonist prefers WhatsApp, apparently with better services.

*#4:* Namiji kamar service ɗin waya yake, yanzu zai kawo yanzu zai ɗauke/A man is like a network service, now you get him, now you don't.

This neoproverb is a reflection of intergender feelings among young Gen Z users of Hausa social media. It alludes to the unpredictability of a steady boyfriend, as it likens a steady relationship to the fluctuating nature of network services, which are very erratic in Nigeria. It is reply to the male proverb stating the unpredictability of a Facebook girlfriend.

## 5. Conclusions

Militz and Militz (1999) remind us that proverbs in actual use are verbal strategies for dealing with social situations. To understand the meaning of proverbs in actual speech acts, they must be viewed as part of the entire communicative performance. Neoproverbs, under the guise of whatever terminology, are clearly referring to a folk proverb, either in semantic structure or in reimagining. I argue that they are neither fractured proverbs (as Wolfgang Mieder would put it), nor disruptive cultural deviants that are radical, parallel compositions (as Aderemi Raji-Oyelade would put it). They do not negate, reverse, or blaspheme. They simply reflect younger Gen Z users of social media creating new proverbs that reflect contemporary social cultures and conversational practices. Regretfully, there

is an insufficient focus on the emergence of new proverbs, for as Wolfgang Mieder (2008, p. 113) noted, "[it] must, however, be said that scholars hitherto have simply not paid enough attention to the creation and emergence of new proverbs" p. 113.

Further, the power of the English language as the internet language in northern Nigeria has created linguistic situations where English or technical words that have no local language equivalents have been incorporated into the idioms, sayings, proverbs, and every day conversations of younger people in the area. Hausa proverbs documented by Merrick, Kirk-Greene, and Yunusa remain the classical references for 'pure' Hausa proverbs. This purity was maintained in them as a result of the closeted nature of Hausa societies before the advent of British colonialism (1903–1960). It is unrealistic to assume that such proverbs would continue to reflect the thinking of a social world confronted with the postcolonial flow of media products, including communication processes curated with information technology. True, as Mieder painfully reminds us, "[m]ost anti-proverbs are one-day-wonders in that they will never enter general folk speech by gaining a certain currency and traditionality." (Mieder 2008, p. 3). Yet, with the increasing adoption of internet technologies as part of every language discourse among both urban and rural Hausa, neoproverbs promise to be part of sustainable, continuing, new folk wisdom in Hausa societies.

Users of such wholly new or modified proverbs may not be aware of the original form of the proverb, and certainly those who create social media proverbs, or those inspired by communication technologies, do so not with the intention of distorting the original proverb, but of simply borrowing the syntactic structure of the traditional proverbs to express their own immersion. Musere (1999, p. 1) notes that "African proverbs involve a wealth of 'disappearing' oral wisdom and tradition that begs for much further exploration". Yet, the continuous creation of new proverbs based on the usage of social media clearly shows a new direction in contemporary paremiology.

**Funding:** This research received no external funding.

**Institutional Review Board Statement:** Not applicable.

**Informed Consent Statement:** Not applicable.

**Data Availability Statement:** The data for this research was harvested from being part of personal conversation groups in the city of Kano as well as few random comments on various Facebook posts. The date is therefore unique and has not been archived on electronic means.

**Conflicts of Interest:** The author declares no conflict of interest.

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
