# Peer review of "“Komai Nisan Dare, Akwai Wani Online”: Social Media and the Emergence of Hausa Neoproverbs"

_humanities, doi:10.3390/h12030044_

Round 1
Reviewer 1 Report
The article under review delves into the enduring relevance of Hausa proverbs in young communities and introduces the concept of neoproverbs. While the author effectively imparts knowledge and engages with existing literature, there are a few areas that require further development and clarification.
The author engages critically with existing literature on the emergence of new proverbs. However, it would be beneficial to expand the discussion beyond African proverbs to provide a more comprehensive perspective on the subject before focusing on the special relevance of Africa regarding the emergence of new proverbs.
The article proposes the concept of neoproverbs, but the need for a new term is not sufficiently justified. A stronger rationale should be provided to clarify how the term contributes to the existing discourse on proverbs.
It remains unclear whether the exclusive focus on technology is deliberate for the purpose of the paper or if neoproverbs are primarily associated with this domain.
The article lacks comprehensive information on the frequency of neoproverbs. Including data in this regard would add depth to the analysis and support the author's arguments.
The provided feedback consists of general comments, and for specific and detailed remarks, please consult the accompanying PDF file.

Reviewer 2 Report
The paper addresses traditional and innovative use of proverbs in the contemporary culture of the Hausa in northern Nigeria. Its main focus is on the modification of traditional proverbs and the emergence of new proverbs in social media, and the way these reflect the thinking of 'Gen Z' online community of Hausa speakers.
The paper is well-structured and balanced. The author clearly states the sources and methodology for gathering his data, from printed, oral, and online sources, representing different ways of proverb usage. The needed amount of social and situational context is provided which is crucial to understanding how and why proverbs are transformed or new proverbs emerge in certain communication situations. The author also provides the context of previous research on the transformation of proverbs, esp. the concept of 'antiproverbs' (Mieder) and 'post-proverbials' (Raji-Oyelade). While considering their work useful for his research, the author finds that "neither Mieder nor Raji-Oyelade considered the status of newly formulated proverbs, rather their interest is on alteration of existing ones" (lines 154-155), and considered that a new term ('neoproverb') is needed "for the category of proverbs that had their origins wholly in internet technologies" (lines 156-157). In my opinion, it could be useful to also consider (and maybe reference) some works which focus not on the transformation of proverbs, but on the constant emergence of new proverbs. Mieder and his colleagues published the Dictionary of Modern Proverbs in 2012, and supplements are regularly published in the journal Proverbium. These collections (and also a number of articles) underline the same fact that is of central importance in the article reviewed here: proverbs by nature reflect, react to, or adapt to the social context. (Cf. in this article: [neoproverbs] "simply reflect younger Gen Z users of social media creating new proverbs that reflect contemporary social cultures and conversational practices" - lines 611-612)
After describing traditional Hausa proverbs and Hausa 'antiproverbs' that are used in contemporary oral communication, 'neoproverbs' are presented as texts that are defined by the place of usage (social media), the users (Gen Z) and the topic of the texts (online communication and connected cultural elements). This is a well-defined cluster of texts that have shared properties and are suitable for analysis on this basis. But I am not sure that the term 'neoproverb' is needed or suitable to define this group. A few questions connected to this could be considered, and maybe some minor revisions could be made to the article, maybe slightly shifting the focus from categorisation and terminology.
First, antiproverbs are often differentiated from 'true' proverbs by the fact that they are playful innovations that have not (yet) gained currency. But they can become proverbs on their own right if they 'catch on' and become more generally known and used (Mieder cites numerous examples in the Dictionary of Modern Proverbs). They are also considered to be based on an already existing proverb or proverb structure.
Neoproverbs in this article are described as a group of texts defined by a) topic - reflecting directly to social media b) originality - "proverbs that often have no antecedents" (line 382). But what does it mean that they have "no antecedent folk-based clause" (403-404)? How can these texts be identified or defined as (neo)proverbs? What does "borrowing the general syntactical structure of a typical proverb" (lines 531-532) mean in the case of Hausa proverbs? I find that a more detailed elaboration of these questions in the article would be a useful addition to the thorough description and explanation of new Hausa proverbs in specific social media context. At the same time, the categorisation and subcategorisation, as well as the separation from the categorisation used by other researchers could be slightly less emphasized.
